# Influence of Surface Characteristics of TiO_2_ Coatings on the Response of Gingival Cells: A Systematic Review of In Vitro Studies

**DOI:** 10.3390/ma16062533

**Published:** 2023-03-22

**Authors:** Nagat Areid, Sini Riivari, Faleh Abushahba, Khalil Shahramian, Timo Närhi

**Affiliations:** 1Department of Prosthetic Dentistry and Stomatognathic Physiology, Institute of Dentistry, University of Turku, FI-20014 Turku, Finland; 2Turku Clinical Biomaterials Center (TCBC), University of Turku, FI-20014 Turku, Finland; 3Oral Health Care, Wellbeing services county of Southwest Finland, P.O. Box 52, FIN-20521 Turku, Finland

**Keywords:** implant abutment, TiO_2_ coatings, nanoporous, human gingival fibroblasts, epithelial cells, surface chemistry, wettability, surface roughness, in vitro

## Abstract

The soft tissue-implant interface requires the formation of epithelium and connective tissue seal to hinder microbial infiltration and prevent epithelial down growth. Nanoporous titanium dioxide (TiO_2_) surface coatings have shown good potential for promoting soft tissue attachment to implant surfaces. However, the impact of their surface properties on the biological response of gingival cells needs further investigation. This systematic review aimed to investigate the cellular behavior of gingival cells on TiO_2_-implant abutment coatings based on in vitro studies. The review was performed to answer the question: “How does the surface characteristic of TiO_2_ coatings influence the gingival cell response in in vitro studies?”. A search in MEDLINE/PubMed and the web of science databases from 1990 to 2022 was performed using keywords. A quality assessment of the studies selected was performed using the SciRAP method. A total of 11 publications were selected from the 289 studies that fulfilled the inclusion criteria. The mean reporting and methodologic quality SciRAP scores were 82.7 ± 6.4/100 and 87 ± 4.2/100, respectively. Within the limitations of this in vitro systematic review, it can be concluded that the TiO_2_ coatings with smooth nano-structured surface topography and good wettability improve gingival cell response compared to non-coated surfaces.

## 1. Introduction

Implant and abutment surfaces are constantly developed to improve the clinical performance of dental implants. Numerous modifications to the implant surfaces have been suggested for establishing sound and stable osseointegration [1]. Implant abutments are usually made of machined polished surfaces to reduce bacterial attachment and biofilm formation. However, there is good evidence that surface modifications of implant abutment may facilitate peri-implant soft tissue attachment without promoting bacterial colonization [2,3,4]. Firm and healthy attachment between the implant abutment and the surrounding soft tissue is essential to protect the underlying tissues from bacterial invasion and to reduce the risk of peri-implantitis [5].

Considering the transgingival nature of dental implants, each implant component comes in contact with different tissues. This poses a unique functional requirement on each implant component to serve the different demands of the respective tissue-implant interfaces. For instance, good osteogenic properties are necessary at the hard tissue-implant interface to enhance osseointegration [6]. In comparison, the soft tissue-implant interface ideally requires the formation of epithelium and connective tissue seal to hinder bacterial infiltration and to prevent epithelial down-growth [7,8,9]. Consequently, different surface modification techniques have been developed. Both subtractive and additive modifications, such as grit-blasting, acid etching, Ti plasma spraying, electrochemical anodic oxidation and laser techniques, have been applied [2,10,11]. These modifications either promote surface bioactivity or create surface structures with different topographies for guided tissue regeneration [2,12,13]. Modifications of implant surface roughness at the nanoscale level have gained more interest. Nanoporous TiO_2_ coatings are one example of nanoscale surface modifications that have shown good potential to promote soft tissue attachment on implant surfaces [14,15]. Different techniques have been proposed to obtain bioactive TiO_2_ coatings on implant surfaces. These include plasma spraying [16,17], anodic oxidation [18], a sol-gel coating method [19] and hydrothermal (HT) treatment [20] (Figure 1). Nanoscale modifications can alter the surface chemistry and topography, aiming at influencing molecular and cellular activities that promote tissue healing at the titanium–tissue interface [21]. Implant abutment materials with optimal surface chemistry, topography, roughness and wettability are the key influencing factors that affect the initial cell responses at the cell-material interface and ultimately promote a bond with the surrounding tissues [22,23]. Although numerous surface modification techniques have been developed to produce bioactive coatings on medical devices, they have not been adopted in wide clinical use. One of the big challenges of coatings, in general, is the fact that the coatings technique ought to be simple, reproducible and financially sensible to be applied on an industrial scale. Furthermore, the technique should facilitate coatings on surfaces with complex shapes and topographies. 

Various in vitro studies have shown that TiO_2_ coatings enhance the gingival cell response [24,25,26]. However, although the coatings produced with different methods share some similarities in their physicochemical composition, the knowledge of the real impact of surface characteristics on soft tissue behavior is unclear. Although in vitro models cannot truly represent the biological process of oral implant and abutment integration, they help understand the fibroblast and epithelial cell behavior on different surfaces, which is a precursor to establishing a soft-tissue attachment. Hence, this systematic review aims to investigate in vitro studies on TiO_2_ surface coatings in the literature and to shed light on the effect of their surface properties on the responses of gingival cells.

## 2. Materials and Methods

### 2.1. Focus Question 

This systematic review was conducted following the guidelines of Transparent Reporting of Systematic Reviews and Meta-Analyses (PRISMA statement) [27], and it was registered in OSF Registries [28]. The proposed focused question was: “How does the surface characteristic of TiO_2_ coating affect the response of the gingival cells in in vitro studies?”. The question was established according to the PICOS (Population–Intervention–Comparison–Outcomes and Setting) strategy: -Population: Fibroblasts, epithelial cells and/or gingival tissue.-Intervention: TiO_2_ coatings with nanofeature characteristics.-Comparison: Non-coated controls.-Outcomes: Gingival cells and tissue response with a qualitative and/or quantitative evaluation.-Setting: In vitro studies.

### 2.2. Eligibility Criteria

In vitro studies investigating fibroblasts and epithelial cell/tissue response to TiO_2_-coated surfaces with nanofeature surface characteristics were eligible for this review. The type of material in the included studies was titanium or zirconia. The principal outcomes, such as cell adhesion, cell proliferation, cell morphology, gene expression and soft tissue-material interface, were included. The articles had to be published in English and have full text available. Any study with a surface modification other than TiO_2_ coating was not included. Studies that evaluate the response of cells other than fibroblasts and epithelial cells were eliminated. Also, studies that used an anodized technique to produce TiO_2_ were excluded. A recent systematic review was conducted to investigate the behavior of gingival cells on anodized titanium surfaces [29]. TiO_2_ coating protocol that does not induce a nano-structured surface was not considered. Studies that solely address mechanical evaluation and in vivo studies evaluating soft tissue responses were excluded. If a selected publication combined in vivo and in vitro studies, only the in vitro aspect was considered.

### 2.3. Information Sources and Search Strategy 

An electronic search for English language studies published between 1990 and 2022 was performed in the following databases (MEDLINE/PubMed and the web of science). Keywords related to TiO_2_ surface were combined with keywords related to soft tissues with AND/OR as Boolean operators, as shown in Table 1. The reference lists of the selected studies were checked to identify any additional studies related to the topic. Furthermore, additional hand-searching of the databases was also conducted.

### 2.4. Selection Process

The electronic research results were imported into Excel to exclude duplicates. A preliminary screening of titles and abstracts was performed by three independent reviewers (NA, SR and KS), and the irrelevant studies were excluded. Then the full texts of the studies meeting the inclusion criteria or those with inadequate data in the title and abstract were obtained and assessed by the same reviewers. Unrelated studies were eliminated, and in case of disagreement, a consensus was resolved by the fifth author (TN). The inter-reviewer reliability was evaluated using kappa coefficients.

### 2.5. Data Extraction and Analysis 

Data were extracted independently by the same three reviewers and tabulated using two Excel spreadsheets designed for this purpose. The first sheet included the author’s name and year of publication, material type, TiO_2_ coating technique and the biological evaluation (analyzed functions, cell line, duration and the number of replicates). The second sheet contained surface characteristics (surface roughness, wettability and morphology) and the biological outcomes of TiO_2_-coated surface compared to the non-coated surface. 

### 2.6. Quality Assessment of Individual Studies 

A quality assessment of the selected studies was performed following the SciRAP method [30]. SciRAP is a web tool that provides criteria for evaluating the reliability and relevance of in vitro studies. The SciRAP criteria are organized under three different tabs in the tool: “Reporting quality”, “Methodological quality” and “Relevance,” with sets of criteria for each one separately. In this review, the “relevance” criteria were not considered because they are associated with toxicity assessment of human health hazards. The reporting and the methodological criteria used in evaluating the selected studies are rated as “fulfilled”, “partially fulfilled” or “not fulfilled”, depending on the descriptions given for each item in the selected studies. The SciRAP score has a value ranging from 0 (all criteria are judged as “not fulfilled”) to 100 (all criteria are judged as “fulfilled”). Four criteria were removed from the reporting quality evaluation (n = 23) and four from the methodological quality evaluation (n = 15) because they did not apply to the in vitro studies included in this review.

## 3. Results

### 3.1. Study Selection

From the initial search, 401 publications were identified (147 articles from PubMed and 254 articles from Web of Science). After removing duplicates (112), 289 articles were evaluated for titles and abstracts. Of these, 44 papers were selected for full-text evaluation, based on the inclusion and exclusion criteria. In addition, three publications were added manually. A total of 11 articles was included in this systematic review. Kappa values for title and full-text overall agreement were 0.73 and 0.83, respectively, indicating a good agreement. Figure 2 shows the PRISMA flowchart with a detailed overview of the search results.

### 3.2. Quality Assessment of the Included Studies

All the included studies defined machined or polished surfaces as a control surface. The cell density and the number of replicates were reported in all the selected studies. Nine studies stated the manufacturer of the tested materials. Samples sterilization was described in eight studies. All the included studies described the tested cell lines/tissue; however, the number of cell passages was stated only in three studies. The mean reporting quality score was 82.7 ± 6.4/100 and the mean methodologic quality score was 87 ± 4.2/100. Table 2 shows the reporting and the methodological quality scores calculated for the included studies. 

### 3.3. Study Characteristics

In this review, quantitative analysis was not feasible due to the differences in methods used and measured outcomes between the included studies. Therefore, only qualitative analysis was performed.

The study design varied in terms of TiO_2_ coating techniques. Five of the selected studies used sol-gel derived TiO_2_ coatings [31,33,35,38,41]; three used HT-induced TiO_2_ coating [32,34,37]; one used the vapor deposition method [39]; one used spray coating technique [36]; and one used peroxotitanium acid solution as a chemical coating method [40].

All the selected studies produced TiO_2_ coatings with a nano-structured surface. Nevertheless, the substrate material used was titanium, titanium alloy and zirconia. The sterilization method was described in eight studies, five of which used autoclaves [31,32,34,39,41]. The remaining three studies used UV light [36,37,40]. Regarding the biological evaluation, two studies were performed on the tissue culture models [32,33], whereas the other studies were conducted on human gingival fibroblasts (HGFs) [37,38,39,40,41] or epithelial cells [31,34,35,36]. Due to the differences in coating techniques and the material used between the studies, results were compiled in specific subgroups. The selected studies and their main characteristics are listed in Table 3. Their main results are specified in Table 4.

#### 3.3.1. Subgroup 1–Sol-Gel Derived TiO_2_ Coatings 

This subgroup includes different sol-gel derived TiO_2_ coatings produced on titanium or zirconia surfaces [31,33,35,38,41]. All studies defined machined or polished surfaces as a control group.

Meretoja et al. [41] used the dip coating method to obtain sol-gel derived TiO_2_ coating (Sa = 0.255 μm with Sa being the mean surface roughness) and evaluated its effect on HGF cell adhesion. The TiO_2_-coated surface showed a stronger cell attachment strength than non-coated surface after 6 h of adhesion, with detachment percentages of 30 ± 3% and 58 ± 4%, respectively. The cell proliferation rate was significantly higher on coated than on non-coated surfaces at different time points. These observations were confirmed with the SEM evaluation. More cells with an elongated shape and extracellular fibrils were observed protruding toward the TiO_2_-coated surface. In contrast, fewer cells with round shapes were found on non-coated surface. Alternatively, Shahramian et al. [38] demonstrated that sol-gel derived TiO_2_ coating on zirconia surface (Sa = 34.20 nm) did not affect HGF proliferation. The proliferation activity was relatively even on coated and non-coated surfaces. The same research group studied the gingival tissue attachment to TiO_2_-coated zirconia and evaluated the strength of this attachment using a porcine gingival tissue culture model [33]. The gingival tissue was more firmly attached to coated than non-coated surfaces. Laminin (Ln-332) was also detected in the epithelial cells adjacent to the coated surface. Riivari et al. [31,35] used human gingival keratinocytes (hGKs) to investigate the behavior of epithelial cells on TiO_2_ coatings derived by in sol polycondensation technique. They showed that the number of attached cells, determined by DNA amount, and the proliferation rate were significantly higher on coated zirconia with a low water contact angle (WCA = 53.0° ± 4.8°) compared to non-coated zirconia. Furthermore, In-sol polycondensation derived TiO_2_ coating enhanced the expression Ln γ2, integrin α6 and integrin β4, which consequently has the potential to promote mucosal attachment on implant surfaces [31]. The confocal image analysis revealed higher signals of Ln γ2, integrin α6 and β4 on TiO_2_-coated surfaces. Meanwhile, vinculin and paxillin were more diffusely in the cytoplasm (Figure 3).

#### 3.3.2. Subgroup 2–HT-Induced TiO_2_ Coatings and Acidic Treatment

This subgroup includes TiO_2_ coatings produced by HT or acidic treatments on titanium surfaces [32,34,37,40]. The included studies considered the machined or polished surfaces as a control group. 

Hoshi et al. [40] developed TiO_2_ coating using a peroxotitanium acid solution. They reported that the TiO_2_ coating (Ra = 228.3 ± 22.1 nm) with anatase structure enhanced surface wettability and promoted fibroblast proliferation compared to the non-coated surface. The cellular response depended upon the coated film thickness since the higher cell number was observed on the coating films with a thickness of 3 µm. Similarly, Areid et al. [37] compared TiO_2_-coated using HT and sol-gel derived techniques with machined non-coated surfaces. The HT-induced TiO_2_-coated surface (WCA 31.1° ± 2.5°) and the sol-gel derived TiO_2_-coated (WCA 35.3° ± 4.3°) showed greater cell attachment strength than non-coated surface, with detachment percentages of 36.4%, 35.8% and 70.7%, respectively. The HGF proliferation rate was improved with time on both coated and non-coated surfaces. These findings were confirmed with SEM images that revealed a thick and uniform cell layer on coated and non-coated surfaces. 

Furthermore, Sakamoto et al. [34] hypothesized that HT treatment would enhance gingival epithelial cells’ adhesion and protein adsorption. The HT-treated titanium surface (Ra = 0.072 ± 0.010 μm) with anatase crystal structure exhibited a hydrophilic surface (WCA = 8.0° ± 1.6°) and showed greater cell adhesion strength than the control group. A higher expression of integrin β4 and a significant amount of Ln-332 adsorption were observed on the HT surface compared with the control surface. Areid et al. [32] evaluated the gingival tissue attachment to HT-induced TiO_2_-coated surface using a pig mandibular block culture model. The microscopic evaluation suggested that the pig tissue explants established soft and hard tissue attachment to coated and non-coated titanium surfaces. The epithelial cells appeared attached closely to the coated surface. In addition, several fibroblasts with some collagen bundles were observed along the HT-coated surface.

#### 3.3.3. Subgroup 3–TiO_2_ Coatings by Spray Coating and Deposition Techniques

This subgroup includes TiO_2_ coating produced by the spray coating technique [36] and deposition coating technique [39]. The machined surface was used as a control group. 

Vignesh et al. [39] observed the growth and attachment of murine fibroblasts on TiO_2_ nanoparticle-coated surface (pits of 1.5 μm depth and 3–5 μm diameter). After 48 h of culture, SEM evaluation showed cells with variable shapes and dimensions and long cytoplasmic extensions on nanoparticle-coated titanium surface. The fibroblast spreading was seen more on the coated titanium surface; meanwhile, no specific cell organization was detected on the machined surface [39]. In contrast, Masa et al. [36] examined the attachment and the proliferation of epithelial cells on nanohybrid-coated titanium surface. After 24 h, no difference was found in cell attachment and proliferation between the polished control (Ra = 0.13 ± 0.01 μm) and TiO_2_-coated surface (Ra = 1.79 ± 0.13 μm). However, the fluorescence images showed a difference in cell morphology. After one week, more polygonal cells with several filopodia were detected on the polished surface. In addition, fewer spread cells with a round shape were found on TiO_2_-modified surface. 

## 4. Discussion

This systematic review evaluated the existing literature regarding TiO_2_ coatings and their effect on gingival cell response in in vitro studies. According to the keywords and eligibility criteria, 11 studies were included in this systematic review. The TiO_2_ coating surface characteristics and their impact on the biological behavior of gingival cells were evaluated in all included studies. Establishing a proper soft tissue bond between the implant abutments and the surrounding soft tissue is essential for successful dental implants [22]. This soft tissue seal around the implant abutment protects the tissue-implant interface from bacterial invasion, which may lead to clinical complications and eventually result in implant loss [42]. It is well-known that the interaction of the implant abutment surfaces with soft tissue is guided mainly by its surface properties, such as surface wettability, surface chemistry and roughness, which have been considered significant factors that affect soft tissue health and stability [3,22].

Nanoscale topography of implant surfaces may affect the surface chemistry and morphology, altering the surface interaction with protein and thus influencing favored cellular response [43]. Vignesh et al. [39] showed that the deposition of TiO_2_ nanoparticles on the titanium surface produced a nanotexture morphology and showed an increased surface oxide composition on the titanium surface. This modified TiO_2_ nanoparticle-coated surface exhibited more fibroblast cell adhesion and spreading than the machined surface, which could indicate the potential of the nanoparticle surface to modulate initial cell attachment response. In addition, the sol-gel derived TiO_2_ coating techniques resulted in smooth, uniform nanoporous TiO_2_ coatings [33,35,38,41]. However, some cracks were apparent in high magnification on the superficial layers of the coated surfaces. In comparison, the HT-induced TiO_2_ surfaces were entirely covered with coating crystals consisting of nearly spherical nanoparticles of 20–50 nm [32,37]. Both sol-gel and HT-induced TiO_2_ surfaces enhanced fibroblast and epithelial cell responses.

The wettability of implant material is a vital surface property that affects the initial cell response at the cell-material interface [44]. WCA measurement is commonly used to determine surface wettability [44]. Studies by Areid et al. [37] and Hoshi et al. [40] demonstrated better wettability for the TiO_2_-coated compared to non-coated surfaces. The higher wettability, which represents hydrophilic surfaces, may partly explain the higher fibroblast cell strength and attachment for coated than non-coated surfaces. Nevertheless, the metabolic activity of HGFs, which represents the proliferation rate, was higher on non-coated surface than those grown on HT-induced TiO_2_ surface on days 7 and 10 [37]. These findings were explained by the fact that the firm attachment of fibroblast cells on the coated surface might slow down the proliferation rate [37]. Moreover, it has been demonstrated that the proliferation activity of fibroblast cells depended upon the TiO_2_-coated film thickness, being better on the thick coating film (3 µm) than on thin coating films (1–2 µm) [40]. The authors explained that the TiO_2_ coating films showed more extensive hydrophilicity than the non-coated surface. The thick coating film maintained the hydrophilicity for a long time, which may promote fibroblast growth and proliferation. Similarly, the HT-coated titanium surface showed greater epithelial cell adhesion strength and higher expression of integrin β4 compared with the non-treated surface [34]. HT treatment also enhanced surface hydrophilicity with minimal change to surface topography and promoted protein adsorption. These surface properties, together with the influence of integrin on cellular function, could contribute to better epithelial cell response and attachment [34].

Sol-gel derived TiO_2_ coatings with different versions improved fibroblast and epithelial cell functions [31,35,38,41]. These observations have been attributed to the smooth surface of the coatings and the ability of TiO_2_ coating to induce calcium phosphate growth on its surface, which is believed to enhance protein adsorption and facilitate cell adhesion, thereby resulting in a better soft tissue attachment as indicated in previous studies conducted by the same research group [19,45,46]. Better cell attachment is often correlated with enhanced adhesion protein expression. Therefore, Riivari et al. [31] evaluated the quality of epithelial cell attachment by detecting the adhesion molecules. They showed that the expression of adhesion proteins was increased for human gingival epithelial cells seeded on TiO_2_-coated surfaces than the non-coated surface [31]. These adhesion proteins were located at hemidesmosomes which may indicate their role in cell adhesion [47]. The expression of paxillin and vinculin levels was also detected on TiO_2_-coated surfaces. Paxillin and vinculin are essential proteins associated with focal adhesion strength. Vinculin is an actin-binding protein believed to play a significant role in integrin-mediated cell adhesion [48]. The authors also reported better surface wettability on the coated than on the non-coated surfaces [35]. These observations were thought to be related to the surface reactivity and topography of the coating, both of which can facilitate protein adsorption and enhance cell attachment to implant surfaces [14,49]. 

In contrast, Masa et al. [36] found no difference in epithelial cell attachment between nanohybrid TiO_2_-coated surface (Ra = 1.79 ± 0.13 μm) and the polished control surface (Ra = 0.13 ± 0.01 μm). The lower cell count and poor proliferation rate results that followed on the TiO_2_-copolymer-modified surface contrast with the literature. One possible explanation indicated by the authors was the sensitivity of primary cells to the polymer matrix or polyacrylate leaching from coatings. These nanohybrid coatings have static biocompatibility as they did not improve cell proliferation, but neither decreased them. The studies included in this review showed that TiO_2_-coated surfaces have positive responses on gingival cells compared to non-coated control surfaces. This result is consistent with a recently published review by Crenn Marie-Joséphine et al. [29], who showed that electrochemically anodized titanium surfaces positively influence gingival cell response compared to conventionally polished or machined surfaces. 

Regarding soft tissue response to TiO_2_-modified surfaces, Shahramian et al. [33] reported that the gingival tissue was more firmly attached to TiO_2_-coated zirconia than non-coated zirconia. The expression of Lnγ2 was also identified at two weeks on the epithelial tissue in contact with the TiO_2_-coated zirconia surface. This result agreed with previous in vivo studies [50,51]. Ln-332 is known to be a crucial molecule for the proper implant-epithelium attachment. The early synthesis and deposition of Ln-332 in the epithelial bond with sol-gel derived TiO_2_-coated zirconia may promote soft tissue attachment. The result indicated that the coated zirconia surface is more attached to the surrounding gingival tissue. In the Areid et al. [32] study, despite the sloughing of the uppermost epithelial cell layers, the epithelium appeared to be in close contact with the coated surface. In addition, CK 14 protein, typically expressed by basal cells of stratified epithelium, was detected in the basal layers of pig gingival epithelium but not in the epithelium close to the implant surface, mimicking the peri-implant epithelium. This finding is consistent with a previous study by Roffel et al. [52], who evaluated the implant-soft tissue interface on a reconstructed human gingiva model. They showed that epithelium adjacent to the titanium abutment showed a specific immunoprofile resembling the peri-implant epithelium [52]. These observations suggested that the TiO_2_ coating seems to have a favorable tissue response. 

Previous in vivo studies showed that the nanoporous TiO_2_ coatings showed better soft tissue outcomes [15,53]. A randomized controlled clinical study by Hall et al. investigated the effect of a nano-structured anodized abutment surface on soft tissue health. They demonstrated that anodized abutments showed a lower bleeding index upon abutment removal and a greater height of keratinized mucosa throughout the 2-year follow-up compared with control abutments. However, no difference in bacterial colonization was observed between the anodized and non-anodized abutments [53]. Wennerberg et al. evaluated the soft tissue attachment and the inflammatory reaction between TiO_2_-coated and non-coated implant abutments in a randomized, comparative clinical study. TiO_2_ surface modification showed more soft tissue adherence, less inflammation and less bone resorption than the control abutments [15]. These in vivo clinical studies supported the results of the previous animal experiments [14,45,46,54] and showed potential clinical benefits to promote soft tissue attachment on implant surfaces.

Although all the included studies in this review produced TiO_2_ coatings with a nano-structured surface, they did not always give a broad overview of all surface properties, making comparisons more difficult. Many selected studies described the surface roughness and the surface wetting behavior determined by contact angle measurements which are considered the key factors that affect the initial cell responses at the cell-material interface and ultimately promote a bond with the surrounding tissue [44]. It is worth noting that surface roughness and wettability are interrelated. The effects of nano-structured surfaces and hydrophilicity on the biological response have been observed but are not well understood [44].

Although all the studies included in this review investigated the cellular behavior of gingival cells on TiO_2_ coatings, surface characteristics reported were inconsistent between studies. In addition, different methods were used to evaluate the gingival cell response, which limits the comparison justification. Nevertheless, these variations allow room for discussion about a possible influence on the results. More studies with a consistent design and methodology will be needed for a thorough evaluation. 

## 5. Conclusions

Based on the findings of this systematic review of in vitro studies, it can be concluded that the TiO_2_ coatings improve fibroblast and epithelial cell responses in terms of adhesion, proliferation and adhesion protein expression compared with non-coated surfaces. The surface chemistry, roughness and wettability may affect the cellular response of gingival cells in different levels depending on topographic characteristics. Smooth surfaces enhance the initial adhesion of fibroblast and epithelial cells. In this regard, HT-induced TiO_2_ and all sol-gel derived TiO_2_ coatings with different versions smooth the surface topography and enhance cell behavior. The TiO_2_-coated surfaces with good surface wettability influence initial cell response at the cell-material interface. Additional modifications prior to TiO_2_ coatings do not seem to promote gingival cell response. 

The observations regarding tissue culture studies suggest that the TiO_2_ coating favors soft tissue attachment; however, care should be taken when interpreting these results since the surface properties and the substrate materials are not similar, creating limitations in correlating the outcomes. Further in vivo studies are needed to prove the real potential of the coating on soft tissue health and maintenance.

## Figures and Tables

**Figure 1 materials-16-02533-f001:**
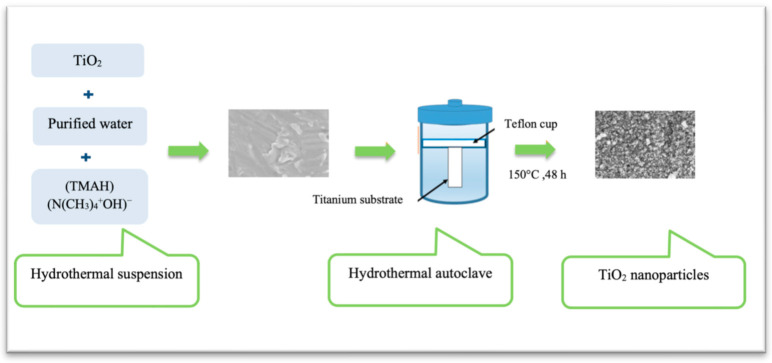
A schematic illustration of the HT treatment to produce nanostructure TiO_2_ coating.

**Figure 2 materials-16-02533-f002:**
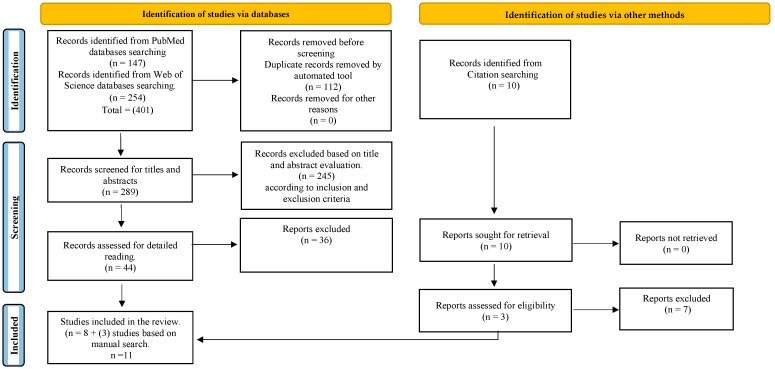
PRISMA flow chart of the study selection process.

**Figure 3 materials-16-02533-f003:**
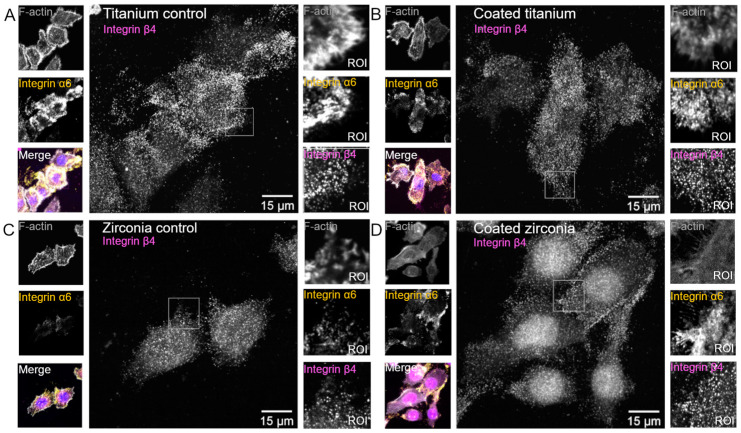
Confocal microscopy images demonstrate the expression of adhesion proteins integrin α6 and β4 on (**A**). non-coated titanium, (**B**). TiO_2_-coated titanium, (**C**). non-coated zirconia and (**D**). TiO_2_-coated zirconia. ROI = reagion of interest (white box). Adapted from [31].

**Table 1 materials-16-02533-t001:** Search strategy and keywords. Asterisk (*) is used as truncation symbol to find variations of search terms.

**#1**	“Dental implant” OR “healing abutment” OR abutment * OR “dental abutment” OR “oral implant” OR “prosthetic abutment”
**#2**	“Titanium dioxide” OR TiO_2_ * OR “titanium oxide” OR titanium dioxide coat * OR titanium oxide coat * OR surface modification* OR modified surface* OR nanotube * OR nanostructure * OR nanoporous *
**#3**	“peri-implant soft tissue” OR gingiva * OR fibroblast * OR “human gingival fibroblast” OR “Gingival epithelial cell” OR keratinocyte * OR mucosa * OR “tissue-implant interface” OR “peri-implant tissue”
**#4**	titanium* OR “Ti6Al4V” OR Zircon *
**#5**	#1 AND #2 AND #3 AND #4

**Table 2 materials-16-02533-t002:** Reporting and methodological quality scores of the included studies using the SciRAP method.

Author (Year)	Reporting Quality	Methodologic Quality	References
**Riivari et al., 2022**	83	90	[31]
**Areid et al., 2021**	85	80	[32]
**Shahramian et al., 2020**	79	88	[33]
**Sakamoto et al., 2019**	87	90	[34]
**Riivari et al., 2019**	88	87.5	[35]
**Masa et al., 2018**	85.5	85	[36]
**Areid et al., 2018**	85.5	88	[37]
**Shahramian. et al., 2017**	68	85	[38]
**Vignesh et al., 2015**	84	86.5	[39]
**Hoshi et al., 2010**	89.5	97	[40]
**Meretoja et al., 2010**	74	85	[41]

**Table 3 materials-16-02533-t003:** Included studies with their major characteristics.

Author/Year	Material type	Tio_2_ Coating Technique	Control Group	Cell Line/Tissue	Analyzed Functions, Methodology, Cell Density, Duration	Number of Replicates
**Riivari et al., 2022 [31]**	Zirconia grade 5 Titanium alloy (Ti-6Al-4V)	A novel in sol TiO_2_-polycondensation coating	Non-coated zirconia Non-coated polished titanium	hGKs from biopsies	Cell adhesion using picoGreen dsDNA assay at 1, 3, 6 and 24 h Cell proliferation by Alamar Blue assay at 1, 3 and 7 d Gene expression of adhesion-related proteins (Ln γ2, ITGα6, ITGβ4, vinculin and paxillin by western blot method at 3 d) Cell spreading, actin cytoskeleton, and focal adhesion proteins by confocal scanning microscopy at 24 h Cell density (25.000 cells/cm^2^) for all experiments	n = 3
**Areid et al., 2021 [32]**	Titanium alloy (Ti-6Al-4V)	HT-induced-TiO_2_ coating	Non-coated polished titanium	Tissue culture model using mandibular pig block including gingival soft tissues.	Tissue-implant interface attachment after histological stainings and analysis by light microscope at 7 and 14 d Immunohistological analysis to detect CK 14 protein after immunohistochemical staining using a light microscope at 14 d	n = 2
**Shahramian et al., 2020 [33]**	Zirconia	Sol-gel derived TiO_2_ coating	Non-coated zirconia	Porcine gingival tissue culture model	Biomechanical measurement analysis by dynamic mechanical analysis 7 and 14 d Immunohistological analysis to identify Ln γ2 chain specific for Ln-332 after immunohistochemical staining using a light microscope at 7 and 14 d	n = 2/time-point
**Sakamoto et al., 2019 [34]**	Titanium alloy (Ti-6Al-4V)	HT-treatment	Non-coated polished titanium	GE1 mouse-derived gingival epithelial cell line	Protein Adsorption Assay (Ln) by fluorescence intensity at 1 h Initial cell attachment by CCK assay (5 × 10^4^ cells/well) at 1 h Cell proliferation by CCK assay (5 × 10^4^ cells/well) at 1, 3 and 7 d Cell adhesion strength by adhesion assay at 1 d Expression of ITGβ4, Nucleus, and Actin Filaments after staining and analysis by fluorescence microscope at 1 d	n = 6
**Riivari et al., 2019 [35]**	Zirconia	Sol-gel derived TiO_2_ coating	Non-coated zirconia	hGKs from biopsies	Cell adhesion using picoGreen dsDNA assay (20.000 cells/cm^2^) at 1, 3, 6 and 24 h Cell proliferation by Alamar Blue assay (25.000 cells/cm^2^) at 1, 3 and 7 d Cell morphology by using a light microscope	n = 4
**Masa et al., 2018 [36]**	Titanium grade IV	TiO_2_ nanohybrid films using spray coating technique	Non-coated polished titanium	Primary HGECs Passage: at least three times	Cell adhesion by MTT assay at 24 h (1 × 10^4^ cells/well) Cell proliferation by MTT assay at 3 and 7 d (1 × 10^4^ cells/well) Cell morphology after staining and by fluorescent microscope at 1, 3 and 7 d	n = 4
**Areid et al., 2018 [37]**	Titanium alloy (Ti-6Al-4V)	HT- induced-TiO_2_ coating	Non-coated polished titanium	HGFs from biopsies Passages: 8 and 10	Cell adhesion resistance against enzymatic detachment by fluorescence microscope at 6 h (20,000 cells/cm^2^) Cell proliferation by Alamar Blue assay (20,000 cells/cm^2^), at 1, 3, 7 and 10 d. Cell morphology by SEM at 6 h and 1, 3, 7 and 10 d	n = 6 n = 4
**Shahramian et al., 2017 [38]**	Zirconia	Sol-gel derived TiO_2_ coating	Non-coated zirconia	HGFs from biopsies	Cell proliferation by Alamar Blue assay (20,000 cells/cm^2^) at 1, 4, 7 and 12 d.	n = 4
**Vignesh et al., 2015 [39]**	Titanium grade II	Pulse laser deposition	Non-coated machined titanium	L929 murine fibroblasts	Cell attachment and growth by SEM (5 × 10^3^ cells/ well) at 48 h	NS
**Hoshi et al., 2010 [40]**	Titanium grade II	Chemical treatment using peroxotitanium acid solution	Non-coated polished titanium	HPLFs from biopsies Passages: 6 and 8	Initial cell spreading and morphology by TMS inverted microscope (1.8 × 10^4^ cells/cm^2^) at 12 h and 3 d Cell proliferation by cell count using haematocytometer (1.8 × 10^4^ cells/cm^2^) at 3 and 7 d	n = 9
**Meretoja et al., 2010 [41]**	Titanium grade II	Sol-gel derived TiO_2_ coating	Non-coated polished titanium	HGFs from biopsies	Initial cell attachment by picoGreen dsDNA assay (25,000 cells/cm^2^) at 1, 3, 6 and 24 h Cell adhesion after serial trypsinization by picoGreen dsDNA assay (12,500 cells/cm^2^) at 6 h Cell proliferation by MTT assay (12,500 cells/cm^2^) at 1, 3, 5, 7 and 10 d Cell morphology by SEM at 6 h, 1, 3 and 7 d and Confocal fluorescence microscopy at 3 d Ultrastructural analysis by TEM at 7 d	n = 4

Abbreviations: HGFs: Human gingival fibroblasts; hGKs: human gingival keratinocytes; HPLFs: Human periodontal ligament fibroblasts; HGECs: Human gingival epithelial cells; HT: Hydrothermal treatment; ITGβ4: integrin β4; ITGα6: integrin α6; Ln γ2: laminin γ2; CCK assay: cell counting kit assay; SEM: scanning electron microscope; TEM: transmission electron microscopy; CK14: cytokeratin 14; MTT assay: dimethylthiazol–diphenyl tetrazolium bromide assay; NS: Not Specified; h: hour; d: day.

**Table 4 materials-16-02533-t004:** Main results of the included studies.

Author/Year	Surface Morphology	Surface Roughness	Water Contact Angle (WCA)	Analyzed Functions and Duration Time	Results Compared to the Control Surface
**Sol-gel derived TiO_2_ coatings**					
**Riivari et al., 2022 [31]**	Smooth surfaces with small, nanostructure particles similar in size and shape on zirconia and titanium-coated surfaces	NS	NS	Cell adhesion at 1, 3, 6 and 24 h	Cell adhesion was greater on TiO_2_-coated zirconia compared to non-coated surface at 24 h.
				Cell proliferation at 1, 3 and 7 d	Cell proliferation was higher on coated zirconia at 1 d and on coated titanium at 3 and 7 d compared to non-coated surfaces.
				Gene expression at 3 d	Significant induction of Ln-γ2 and ITG α6 on coated surfaces. ITG β4 was higher on coated titanium. Paxillin and vinculin levels were higher on the coated zirconia.
				Cell spreading, actin cytoskeleton and focal adhesion proteins at 24 h	Cell spreading was higher on both coated surfaces. The expression of Ln-γ2, ITG α6 and ITG β4 was higher on the coated samples.
**Shahramian et al., 2020 [33]**	Smooth surface with some cracks on the superficial layer of coating (Data from Shahramian et al., 2017, that used the same material)	TiO_2_-coated zirconia: Sa = 34.2 nm Control: Sa = 533.8 nm. (Data from Shahramian et al., 2017, that used the same material)	TiO_2_-coated zirconia: 53.0° ± 4.8° Non-coated: 74.1° ± 6.9° (data from their previous work, Riivari et al., 2019)	Biomechanical analysis at 7 and 14 d	Tissue attachment to coated zirconia demonstrated higher dynamic modulus of elasticity and higher creep modulus than non-coated zirconia at 7 and 14 d.
				Gene expression at 7 and 14 d	At 7 d Ln-γ2 was detected at the epithelial cells adjacent to the coated surface but not on the non-coated surface. At 14 d Ln-γ2 was strongly expressed along the innermost layer of the epithelium with the coated surface.
**Riivari et al., 2019 [35]**	NS	NS	TiO_2_-coated zirconia: 53.0° ± 4.8° Non-coated: 74.1° ± 6.9°	Cell adhesion at 1, 3, 6 and 24 h	Cell adhesion was higher on TiO_2_-coated zirconia compared to non-coated surface at 24 h.
				Cell proliferation at 1, 3 and 7 d	Cell proliferation on TiO_2_-coated zirconia was higher compared to non-coated surface at 3 and 7 d
				Cell morphology by SEM	More cells with more uniform cell layers on coated zirconia
**Shahramian et al., 2017 [38]**	Smooth surface with some cracks on the superficial layer of coating	TiO_2_-coated zirconia: Sa = 34.2 nm Control: Sa = 533.8 nm	TiO_2_-coated zirconia: 53.0° ± 4.8° Non-coated: 74.1° ± 6.9°	Cell proliferation at 1, 4, 7 and 12 d.	Cell proliferation was higher on the coated surface at d 1 and lower at d 4. The proliferation activity was relatively even on both surfaces at 7 and 12 d.
**Meretoja et al., 2010 [41]**	Uniform surface with extensive cracking	Sol-gel derived TiO_2_ coated surface: Sa = 0.255 μm Sq = 0.322 μm	NS	Initial cell attachment at 1, 3, 6 and 24 h	No difference cells were found between the coated and non-coated surfaces.
				Cell adhesion at 6 h	Stronger cell attachment strength was observed on the TiO_2_-coated surface.
				Cell proliferation at 1, 3, 5, 7 and 10 d	The proliferation rate was higher on coated surface than on non-coated surface at all time points.
				Cell morphology at 6 h, 1, 3 and 7 d and Confocal fluorescence microscopy at 3 d	More elongated cells are attached on TiO_2_-coated surface, with extracellular fibrils protruding towards the coated surface. Cells were largely increased at d 3, and cell mass with uniform thickness was formed at d 7. Well-organized actin cytoskeleton and focal contacts were observed on both surfaces.
				Ultrastructural analysis at 7 d	A continuous layer of two to three cells thickness covered both surfaces.
**HT-induced TiO_2_ coatings and acidic treatment**					
**Areid et al., 2021 [32]**	Titanium nanoparticles with a diameter of 20–50 nm (Data from Areid et al., 2018, that used the same material)	NS	HT-induced TiO_2_: 31.1° ± 2.5° NC control: 50.3° ± 4.5° (Data from Areid et al., 2018, that used the same material)	Histological analysis at 7 and 14 d	The pig tissue explants showed epithelial and connective tissue appeared attached to both surfaces at 7 d. Several fibroblasts were observed along the coated surface at 7 d. Epithelial cells appeared attached closely to coated surface at 14 d. Some collagen bundles running parallel or slightly oblique to the coated surface at 14 d.
				Gene expression at 14 d	CK14 positivity in the basal cell layer of stratified gingival epithelium. Faint CK14 positivity was detected in the innermost cells facing the coated implant surface.
**Sakamoto et al., 2019 [34]**	HT treatment changed the surface crystal structure into an anatase type of TiO_2_ without an apparent change in surface topography.	HT treated: Ra = 0.072 ± 0.010 μm Rt = 0.95 ± 0.24 μm Control: Ra = 0.070 ± 0.008 μm Rt = 1.03 ± 0.23 μm	HT treated: 8.0° ± 1.6° Control: 78.9° ± 4.8°	Amount of Adsorbed Ln at1 h	The amount of adsorbed Ln was greater on the HT surface than that on the control surface.
				Initial cell attachment at 1 h	No difference
				Cell proliferation at 1, 3 and 7 d	The proliferation activity was lower on the HT surface at 1 and 3 d. No difference was found between the surfaces at 7 d.
				Cell adhesion strength at 1 d	The cell adhesion ratio was greater on the HT surface at 1 d.
				Gene Expression at 1 d	A stronger signal of ITGβ4 was observed on HT coated surface.
**Areid et al., 2018 [37]**	Nanoparticles with a diameter of 20–50 nm	NS	HT-induced TiO_2_: 31.1° ± 2.5° NC control: 50.3° ± 4.5° Sol-gel derived TiO_2_: 35.3° ± 4.3°	Cell adhesion resistance against enzymatic detachment at 6 h	The detachment percentage was lower on coated surfaces than on non-coated surface.
				Cell proliferation at 1, 3, 7 and 10 d.	The proliferation rate improved with time on both surfaces. Lower cell activity showed on the coated surface at 7 and 10 d.
				Cell morphology at 6 h and 1, 3, 7 and 10 d	More cells with an elongated shape were observed on coated surfaces.
**Hoshi et al., 2010 [40]**	Smooth surface texture with anatase structure	TiO_2_-coated 228.3 ± 22.1 nm non-coated 275.7 ± 23.5 nm	(Diagram without associated values) TiO_2_-coated: Hydrophilic or super-hydrophilic Non-coated: Hydrophobic	Initial cell spreading and morphology at 12 h and 3 d	The cell proliferation improved at 12 h and 3 d on TiO_2_-coated surface compared to non-coated surface. Sufficient cellular bridges and proliferation were observed on TiO_2_-coated surface.
				Cell proliferation 3 and 7 d	Fibroblast cells were higher on TiO_2_-coated surface and depended upon the coated film thickness.
**TiO_2_ coatings by spray coating and deposition techniques**					
**Masa et al., 2018 [36]**	TiO_2_-coated showed an amorphous surface pattern. Characteristic grains appeared on the silver-containing coated surfaces.	TiO_2_: Ra = 1.79 ± 0.13 μm polished Ra = 0.13 ± 0.01 μm	NS	Cell adhesion at 24 h	No difference was found between the polished control and TiO_2_-coated surfaces.
				Cell proliferation at 3 and 7 d	No difference
				Cell morphology at 1, 3 and 7 d	At 3 d, well-spread cell morphology was observed on polished surface. At 3 and 7 d, a few poorly spread cells were detected on the coated surface. At 7 d, a more polygonal cell with several filopodia was detected on the polished surface.
**Vignesh et al., 2015 [39]**	Spherical nanoparticles with 20 nm covered with pits 1.5 μm depth 3–5 μm diameter	NS	NS	Cell attachment and growth at 48 h	TiO_2_ nanoparticle-coated surface showed better cell response and attachment than the control group.

Abbreviations: NS: Not Specified; HT: hydrothermal treatment; Ra: average roughness; Rt: maximum roughness height; Sa: arithmetical mean height; Sq: root-mean-square height; NC: non-coated; UV: Ultraviolet.

## Data Availability

All the data is available in the main text.

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
