# Peer review of "Influence of Surface Characteristics of TiO2 Coatings on the Response of Gingival Cells: A Systematic Review of In Vitro Studies"

_materials, 2023, doi:10.3390/ma16062533_

Round 1

Reviewer 1 Report

The authors proposed a systematic review on the In Vitro studies in the litterature and the effect of the surface properties of TiO2 coating on the cellular response on gingival cells. After careful review of this manuscript, I can not recommend the publication of this work in its current form. 

1/ The English language should be improved. 

2/ The aim of the work should be highlighted. The design of the review is not attractive and should be improved.

3 / The authors focused on the TiO2 coating. Why? The various methods for the synthesis of TiO2 coating should be provided. 

4/ The text is not well structured. Many tables were provided but not well informative. 

5/ More shematic illustrations and figures from previous works should be added to be clear for the readers.

6/ The abreviations used in the whole manuscript should be defined when it was used for the first time.

7/ The references should be updated and old references should be replaced by recent ones.

Reviewer 2 Report

Dear authors, 

The study entitled "Influence of surface characteristics of TiO2 coatings on the response of gingival cells: A Systematic Review of In Vitro Studies" is showing a systemic review of modified surfaces (TiO2 coatings) and their responses on soft tissue cells for application on dental abutments.  Despite the very well-performed application of guidelines for systematic reviews and the good structuration of this review, the study presents several faults in terms of merit for publication. A systemic review of in vitro studies should include studies that present quite similar characteristics and analyses because in vitro studies are easy to control and standardize, differently from clinical studies that have several external factors to cause bias. Moreover, studies that present similar final aims. The studies selected in this review are very different, starting with the need of the authors to divide the results regarding surface properties and biological responses. Almost all the studies showed different methodologies for surface treatment, different base materials (Zr, Ti, alloys), and different surface properties, resulting in different surfaces that will affect the cells in different levels of response. 

This comparison is very wide and unfair. 

A second huge issue is that the authors do not show what kind of "control surface" the studies are comparing the promising results. Some studies may compare against a machined surface and other may compare against micro-texturized surfaces. Additionally, control surfaces in different base materials may have better results than other materials and again the comparison becomes unfair.

Even same these studies present in the end a TiO2 coating, they present several differences in properties and materials. The comparison should be made with studies showing the same base materials, similar roughness, and similar wettability. Also, select only studies that compare the results to a control surface with similar characteristics between the studies selected. 

Based on these points cited above, the question "How does the surface characteristic of TiO2 coating affect the response of the gingival cells in in vitro studies?" can not be answered, because the authors selected studies presenting different surface characteristics that can not be related. The readers already know that modified surfaces (micro or nano-scaled) present better responses than control groups, however, the main point to be discussed is why? and what surface property? Responding to this point, this review would be very interesting. 

My suggestion is to the authors change this review to a narrative review and expose their findings with more discussion and more references, exploring this topic in-deep in a narrative way. 

Reviewer 3 Report

Dear authors,

Firstly, congratulations for a great manuscript and extraordinary clinical work. The manuscript titled: "Influence of surface characteristics of TiO2 coatings on the response of gingival cells: A Systematic Review of In Vitro Studies" is very innovative and is backed up with promising results for further clinical studies. Abstract and introduction are well concised and focused, defining main problem of cellular behavior of gingival cells on TiO2-implant abutment coatings based on in vitro studies. Materials and methods are explained in detail and discussion is excellent presented.

Only suggestion is to check some minor mistakes in grammar and text editing.

The whole manuscript could be of great interest to the readership of the journal.

Reviewer 4 Report

Introduction:  

Line 47: '' Some of these surface modifications are grit-blasting, acid etching, Ti plasma spraying, electrochemical anodic oxidation, micro-grooving, and laser techniques. These methods promote surface bioactivity or create surface structures for guided tissue regeneration, resulting in implants with different surface topographies [9]. '' and line 55: '' These include plasma spraying, a sol-gel coating 55 method, anodic oxidation, and hydrothermal treatment. ''  These sentences describe a big range of procedures, but no references are currently cited to give a potential reader the possibility to deepening the knowledges on the mentioned procedures. I suggest the authors to improve the references describing the procedure.

In general, I encourage the Authors to add a little paragraph describing what are the main limitations of the currently existing procedures and why the research on the topic is still open. This will help the fluency of the introduction and will help the readers to understand the originality and rational of your study. 

For both the above points I suggest you a reference on the topic and I encourage you to cite it to help references improvements: 

Carossa, M.; Cavagnetto, D.; Mancini, F.; Mosca Balma, A.; Mussano, F. Plasma of Argon Treatment of the Implant Surface, Systematic Review of In Vitro Studies. Biomolecules 202212, 1219. https://doi.org/10.3390/biom12091219

Results: the results section is a little bit too long, considering that the major findings of each article are also described in the tables. Please try to shorten it.

Discussion: 

to complete your review, please add the state of the art on the in vivo application of the Tio2 currently described in the literature, if any. 

Round 2

Reviewer 1 Report

The paper can be accepted for publication. The authors are only advised to carefully improve the English language and correct some typos and grammatical errors throughout the manuscript.

Author Response

Reviewer 1,

The paper can be accepted for publication. The authors are only advised to carefully improve the English language and correct some typos and grammatical errors throughout the manuscript.

Response 1: The English language is improved, and the typos and grammatical errors were corrected as requested.

Reviewer 2 Report

Dear authors, 

The authors have improved the manuscript with figures, references, and more discussion. However, the main study continues with the same focus without any huge changes. If the authors would like to maintain this systematic review for publication, the conclusion section should be intensely improved to describe exactly what the authors found.  

1- As described in the author's response, the study that compares micro-pit (microtexturization) to nanotexturization should be excluded from this review. This control comparison can not be related to all the other studies. The outcomes from this study demonstrated totally different comparisons, even, the final conclusion is promising for TiO2 nano-coatings. Tables and search designs should be corrected.  

2- Conclusion section: The authors need to expose all the limitations of this study in the conclusions section. 

Firstly: line 469: "The surface chemistry, roughness, and wettability affect the cellular response of gingival cells" 

 "The surface chemistry, roughness, and wettability  "may" affect the cellular response of gingival cells in different levels depending on topographic characteristics." 

- Line: 476: " The observations regarding tissue culture studies suggest that the TiO2 coating favors soft tissue attachment. However, this study should be interpreted carefully, the studies selected do not show totally similar surface properties and do not show the same base material (Zr,Ti,alloys), therefore, generating difficulty in clearly correlating the final outcomes. In vivo studies ...."

These are suggestions for the conclusion section, however, the authors must expose this exact idea of limitation, or I can not support this publication. 

Author Response

Reviewer 2

The authors are grateful for your critical comments and suggestions that have greatly improved the quality of our manuscript.

1- As described in the author's response, the study that compares micro-pit (microtexturization) to nanotexturization should be excluded from this review. This control comparison can not be related to all the other studies. The outcomes from this study demonstrated totally different comparisons, even, the final conclusion is promising for TiO2 nano-coatings. Tables and search designs should be corrected. 

 Response 1:  Kubo et al. study is excluded from this review as requested.  The authors adjusted tables, search designs, results, and discussion.

2- Conclusion section: The authors need to expose all the limitations of this study in the conclusions section. 

Firstly: line 469: "The surface chemistry, roughness, and wettability affect the cellular response of gingival cells" 

 "The surface chemistry, roughness, and wettability  "may" affect the cellular response of gingival cells in different levels depending on topographic characteristics." 

- Line: 476: " The observations regarding tissue culture studies suggest that the TiO2 coating favors soft tissue attachment. However, this study should be interpreted carefully, the studies selected do not show totally similar surface properties and do not show the same base material (Zr,Ti,alloys), therefore, generating difficulty in clearly correlating the final outcomes. In vivo studies ...."

These are suggestions for the conclusion section, however, the authors must expose this exact idea of limitation, or I can not support this publication. 

 Response 2: The limitations were highlighted in the conclusion section, as requested.

Reviewer 4 Report

The Authors modified the manuscript correctly. 

Please change reference 42 with the following doi: 10.3390/ma16010246 which is more related to the topic of your article.

Author Response

Reviewer 4,

Please change reference 42 with the following doi: 10.3390/ma16010246 which is more related to the topic of your article.

 Response 1: The reference 42 is changed with the suggested one (it takes no. 41 in the reference list)